# Monocarboxylate Transporters 1 and 4 and MTCO1 in Gastric Cancer

**DOI:** 10.3390/cancers13092142

**Published:** 2021-04-29

**Authors:** Maarit Eskuri, Niko Kemi, Joonas H. Kauppila

**Affiliations:** 1Cancer and Translational Medicine Research Unit, Medical Research Center, University of Oulu and Oulu University Hospital, 90014 Oulu, Finland; niko.kemi@oulu.fi; 2Surgery Research Unit, Medical Research Center, University of Oulu and Oulu University Hospital, 90014 Oulu, Finland; joonas.kauppila@oulu.fi; 3Upper Gastrointestinal Surgery, Department of Molecular Medicine and Surgery, Karolinska Institutet and Karolinska University Hospital, 17177 Stockholm, Sweden

**Keywords:** monocarboxylate transporter, MTCO1, gastric cancer, immunohistochemistry

## Abstract

**Simple Summary:**

The expression of monocarboxylate transporters (MCTs) are reported in a variety of cancers and suggested as a therapeutic target for cancer treatment. However, previous study results in gastric cancer are contradictory. In this study, we evaluated the expression of MCT1, MCT4, and Mitochondrial cytochrome c oxidase (MTCO1) and their association with clinicopathological parameters and prognostic significance in a cohort of 568 surgically treated gastric cancer patients. The results suggest that monocarboxylate transporters and MTCO1 are associated with gastric cancer progression but have no independent prognostic relevance.

**Abstract:**

*Background:* Monocarboxylate transporters (MCTs) appear to play an important role in tumor development and aggressiveness. The present study aimed to evaluate associations between cytoplasmic MCT1, MCT4, and mitochondrial cytochrome c oxidase (MTCO1) expression and clinicopathological variables or survival in gastric cancer. *Material and methods*: A total of 568 gastric adenocarcinoma patients were included in this retrospective cohort study. Protein expressions were detected by immunohistochemical staining. The patients were divided into low expression and high expression groups by median value. The Chi-squared test was used to compare categorical variables. The T-test was used to compare continuous variables. Expressions were analyzed in relation to 5-year survival and overall survival. Cox regression provided HRs and 95% CIs, adjusted for confounders. *Results:* High cytoplasmic MCT1 expression was associated statistically significantly with higher T-class (*p* = 0.020). High cytoplasmic MCT4 expression was associated statistically significantly with positive lymph node status (*p* = 0.005) and was more common in Lauren’s intestinal type (*p* < 0.001). Low cytoplasmic MTCO1 expression was associated statistically significantly with positive distant metastases (*p* = 0.030), and high cytoplasmic MTCO1 expression was associated more often with intestinal type (*p* = 0.044). However, MCT1, MCT4, and MTCO1 were not associated with survival. *Conclusions:* Monocarboxylate receptors seem to be associated with gastric cancer progression but have no independent prognostic relevance.

## 1. Introduction

Gastric cancer (GC) is the fifth most common cancer and the third leading cause of cancer deaths worldwide [1,2]. In 2020, more than one million new cases were diagnosed [1]. GC is usually discovered in advanced stages with an estimated 5-year survival rate of less than 25% [3]. Tumor staging is commonly used for prognostication, but prognosis is hard to estimate in individual patients. Therefore, new predictive biomarkers are needed.

Normal cells produce energy through aerobic mitochondrial metabolism, while in rapidly growing cancer cells, energy is produced by anaerobic glycolysis, resulting in high levels of lactate and carboxylic acids [4]. To avoid intracellular acidification, lactate is exported to the extracellular space. This pH regulation is controlled by monocarboxylate transporters (MCTs) [4,5]. The activation of glycolysis promotes aggressive proliferation, invasion, and metastatic behavior [5]. MCTs also play a significant role by induction of drug resistance [6,7]. MCTs have recently been suggested as potential therapeutic targets for treating malignant solid tumors [8], and MCT1 inhibitor is currently in phase I clinical trials for cancer treatment [9]. The expression of MCT1 and MCT4 have previously been studied in GC [3,4,7,10,11] as well as many other cancers [12,13,14,15,16]. So far, previous studies in GC are contradictory, and the utility of MCTs in GC is poorly understood. However, one study suggested that MCT1 inhibitor increases the chemotherapy sensitivity in GC, implying that targeting MCT1 in cancer may have therapeutic potential in GC [7]. However, further studies on MCTs in GC are needed for therapeutic applications. Mitochondrial cytochrome c oxidase 1 (MTCO1) belongs to the cytochrome c family, and it is involved in cellular energy production and in cell apoptosis. Previous studies have shown that the expression of MTCO1 is altered in different tumors [17,18]. 

The aim of this study was to evaluate the expression of MCT1, MCT4, and MTCO1 and their association with clinicopathological parameters and prognostic significance in a cohort of gastric cancer patients. 

## 2. Material and Methods

### 2.1. Study Design

This study was a retrospective cohort study in a single institution in a tertiary care hospital in Northern Finland. There were 601 consecutive patients who underwent gastrectomy for gastric cancer in Oulu University Hospital between the years 1983 and 2016. The final series consisted of 568 adenocarcinoma patients whose tissue samples were available for construction of the tissue microarray. 

### 2.2. Data Collection 

The patients were identified from the archives of the Department of Pathology at the Oulu University Hospital, Finland. Clinical data for each patient were obtained from patient records, including operation charts and pathology reports. The immutable national personal numbers assigned to each resident in the country were used to combine data from the patient records and the 100% complete follow-up data from the Causes of Death Registry at Statistics Finland. Follow-up data were available until the end of 2016.

### 2.3. Tissue Microarray

Representative tissue samples with deepest tumor invasion were identified based on diagnostic hematoxylin–eosin slides. The slides were scanned using Aperio AT2 (Leica Biosystems, Wetzlar, Germany). Representative areas of tumor front and tumor center were selected from the scanned slides. Two cores from the tumor front and two cores from the tumor center were taken from each patient tissue block to avoid loss of participants during the experiments and to achieve representative samplings from different parts of the tumor. The cores were punched from paraffin-embedded tissue blocks and transferred to a receiver block, which were used for further staining analysis and construct tissue microarray (TMA). Computer-driven TMA-device Galileo TMA CK4500 (Integrated Systems Engineering, Milan, Italy) was used for construction.

### 2.4. Immunohistochemistry

MCT1, MCT4, and MTCO1 protein expression were detected by immunohistochemical (IHC) staining. TMAs were cut in 4 µm slices, placed on glass slides, deparaffinized in xylene, and rehydrated through graded alcohols. Rehydrated samples were submitted into a microwave oven for antigen retrieval with citrate buffer 800 W for 2 min and 150 W for 10 min and then cooled to room temperature for 20 min. Samples were rinsed in phosphate-buffered saline with Tween (PBS-T), and then, endogenous peroxidase was neutralized in peroxidase blocking solution (Dako S2023) for 5 min. After a wash in PBS-T, sections were incubated with antibodies (Dako S2022); MCT1 (diluted 1:100, Santa Cruz (H-70)), MCT4 (diluted 1:200, Santa Cruz (H-90)), and MTCO1 (diluted 1:500, Abcam). After another wash in PBS-T, samples were incubated with En-vision polymer (Dako K5007) for 3–5 min. After the final wash in PBS-T, Diaminobenzidine (Dako basic DAB-kit) was used as a chromogen. Lastly, the samples were counterstained in hematoxylin for 1 min. All staining was done with Dako Autostainer (Dako, Copenhagen, Denmark). Cancer tissues with high expression of MCT1, MCT4, and MTCO1 were used as an external positive control. 

### 2.5. Assessment of Immunostaining

Sections were scanned and digitized using Aperio AT2 (Leica Biosystems, Wetzlar, Germany). Cores were analyzed from scanned slides using QuPath [19]. Two independent researchers (M.E. and N.K.) who were blinded to the clinical and outcome data performed the analyses. It was decided a priori that the cores for each staining would be analyzed by one researcher (M.E.) if good interobserver agreement, as indicated by a kappa value of at least 0.7 could be achieved in assessment of a sample of 100 cores from randomly selected cases.

We assessed the intensity of staining from 0 (absent) to 3 (strong intensity) and the percentage of positive tumor cells (0–100%) for each core. The mean value of the independent estimates was used for statistical analyses. The mean intensity and mean percentage of assessable cores for each patient cores were used to obtain a histoscore for staining intensity, which was calculated by multiplying the mean intensity and the mean percentage of the cancer cells (values 0–300). For statistical evaluation, the intensity distribution for each stain was dichotomized by median value into two equal-sized groups (low and high).

### 2.6. Statistical Analysis

All statistical analyses were performed using the IBM SPSS Statistics 26.0. (IBM Corp., Armonk, NY, USA). Cohen’s kappa was calculated to analyze interobserver agreement. The Chi-squared test was used to compare categorical variables. The T-test was used to compare continuous variables. The Kaplan–Meier method was used to obtain Kaplan–Meier curves. A Cox regression model was used to perform univariate and multivariable analysis, providing hazard ratios (HR) with 95% confidence intervals (CI). Cox regression was adjusted for potential confounding variables: (1) year of surgery (<2000 or ≥2000), (2) age at diagnosis (continuous variable), (3) sex (male or female), (4) administration of preoperative chemotherapy (yes or no), (5) tumor stage (stage I–II or stage III–IV), (6) Lauren classification (intestinal, diffuse, or mixed), and (7) radical resection (R_0_ or R_1/2_). R_o_ resection was defined as no cancer cells seen microscopically at the tumor border. R_1/2_ resection was defined as tumor growth on the border of the resected specimen, or macroscopic residual disease. *p* values less than 0.05 were accepted as statistically significant. Sensitivity analysis including only curatively intended R_0_ resected patients to exclude any bias from the inclusion of patients with dismal prognosis. As the results of the sensitivity analysis did not differ from main analysis, only results from the main analysis are presented.

## 3. Results

### 3.1. Patients

There were 568 patients included in this study. The median age of the cancer patients was 69 years (range 27–91). A total of 347 (61.1%) patients were men, and 221 (38.9%) were women. The median follow-up time was 26 months (range 0–396). 

Of these 568 patients, 424 (74.6.0%) underwent microscopically confirmed R_0_ resection, and 144 (25.4%) had R_1/2_ resection. The patients with R_1/2_ resection included patients with non-curative intent, as well as 33 (5.8%) patients that had distant metastases at the time of surgery. Only 22 (3.9%) of patients underwent perioperative chemotherapy.

### 3.2. Assessment of MCT1, MCT4, and MTC01 Staining

Of 568 patients, 560 were included in MCT1 staining analysis, 558 were included in MCT4, and 562 were included in MTCO1, as they had at least one assessable core of each staining available. If a core was incomplete or clearly defectively stained, it was excluded from the analysis. All available cores were analyzed, up to four cores from each patient. Each core was evaluated individually.

MCT1, MCT4, and MTCO1 were all expressed in gastric cancer. Staining was mainly cytoplasmic and occasionally focused on membranes. Representative images of immunostainings are shown in Figure 1.

A kappa value of 0.7 was achieved only for MCT1, for which all of the cores were then analyzed by only M.E. The kappa value was 0.4 for MCT4 and 0.6 for MTCO1, and subsequently, both M.E. and N.K. analyzed MCT4 and MTCO1.

The assessment of MCT1 and the mean values of the assessments of the two researchers for MCT4 and MCTO1 were calculated for each core and used for further analysis. Mean values for staining intensity and percentage of stained cells of assessable cores for each patient was calculated, and the histoscore ranging from 0 to 300 was obtained. Then, the patients were divided into low expression and high expression groups by median histoscore value. After the assessment, the median value of MCT1 was 150, 305 (55%) of the patients displayed low expression, and 255 (45%) exhibited high expression. The median value of MCT4 was 25; 290 (52%) of the patients displayed low expression, and 268 (48%) exhibited high expression. The median value of MTCO1 was 188, 299 (53%) of the patients displayed low expression, and 263 (47%) exhibited high expression.

### 3.3. MCT1, MCT4, and MTCO1 Expression Associations with Clinicopathological Variables and Cancer Survival

High cytoplasmic MCT1 expression was associated statistically significantly with higher T-class (*p* = 0.020), while it was not associated with the other factors (Table 1). High cytoplasmic MCT4 expression was associated statistically significantly with positive lymph node status (*p* = 0.005) and Lauren’s intestinal type histology (*p* < 0.001), but not the other clinicopathological factors (Table 2). Low cytoplasmic MTCO1 expression was associated statistically significantly with positive distant metastases (*p* = 0.030) and diffuse type histology (*p* = 0.044) (Table 3). 

The expression of MCT1, MCT4, and MTCO1 was not associated with survival in uni- or multivariate analysis (Table 4). The results of the sensitivity analysis were similar to the main analysis.

## 4. Discussion

In this study, we characterized MCT1, MCT4, and MTCO1 expressions in gastric adenocarcinoma. All three markers were expressed in gastric cancer and associated with some known prognostic factors in gastric cancer but had no independent prognostic relevance. 

There are some strengths and limitations that should be considered before interpreting the results. The strengths of the study include the large size of the study and the lack of selection bias. The retrospective single-institution design might limit its applicability for larger populations. Nevertheless, this study is larger than any of the previous studies on the topic [3,4,7,10,11]. Patients with unradical resections were also included to minimize selection bias and maximize the power of this study. On the other hand, sensitivity analysis excluding palliative and non-radically resected patients showed similar results to the main analysis. The assessment of staining was at times challenging to replicate, as seen in the kappa value of 0.6 in MCT4, indicating moderate agreement and 0.4 in MTCO1 indicating fair agreement, which might limit its applicability in clinical practice. The use of only immunohistochemistry is a possible weakness. Previous studies also assessed these markers in the tumor stroma, while we focused on the cytoplasm of the tumor cells. The present study did not relate the protein expression to chemotherapy sensitivity or protein function in cells, and therefore, this study cannot exclude that MCT1, MCT4, or MTCO1 could have clinical relevance. 

Normal cells produce energy through mitochondrial oxidative phosphorylation. The last product of the glycolysis is pyruvate, which enters the mitochondria where it is oxidized by the citric acid cycle to generate energy [20]. However, in cancer cells, pyruvate is fermented into lactate, even in aerobic conditions and fully functioning mitochondria available. This aerobic glycolysis is called the Warburg effect [21]. To avoid intracellular acidification, lactate must be expelled out of the cell, causing acidification of the extracellular space. Lactate is not only a waste for tumors; cancer cells also become more aggressive and resistant to therapy by acidifying their microenvironment. Low pH and lactate enable cancer cells to migrate, invade, promote angiogenesis, immune escape, and radioresistance [20,21,22]. The MCT family includes 14 members, among which MCT1-4 are catalyzing the proton-linked transport of pyruvate, lactate, and ketone bodies across the cell membrane [23]. MCTs play a significant role in the upregulation of glycolysis and adaptation to acidosis [4]. MCTs might be therapeutic targets for disrupting cancer cell energy metabolism and for starving cancer cells and are therefore interesting targets for drug development [7,21,24]. However, turning back to the original findings of tumor metabolism, targeting both aerobic glycolysis and mitochondrial metabolism may be needed. MTCO1 is a terminal complex of the respiratory electron transport chain of mitochondria [5], and it is involved in apoptosis, for example through the activation of the caspase cascade. If MTCO1 is downregulated, it cannot activate the caspase-dependent cell-death pathway and therefore cannot induce apoptosis [25]. In addition, the generation of reactive oxygen species (ROS) is often causative for programmed cell death. Decreased usage of the mitochondrial respiratory chain results in reduced production of ROS, which promotes cancer cell proliferation and apoptosis evasion [26]. In keeping with the Warburg hypothesis, producing energy through anaerobic glycolysis and altered mitochondrial metabolism are hallmarks of many proliferating tumors [21]. In addition, the relationship between cytochrome c oxidase and chemo- or radioresistance has been reported in different tumor cell lines. A Japanese study reported that the downregulation of MTCO1 induced radioresistance in esophageal squamous cell carcinoma [25].

The role of MCT1 in GC has been evaluated in three previous studies with smaller sample sizes. A Chinese study (*n* = 120) used both immunohistochemistry and PCR in their study. They reported that MCT1 was highly expressed in GC. Higher expression of MCT1 was positively associated with increased overall survival and with advanced TNM stage [7]. A Portuguese study (*n* = 190) with immunohistochemically stained gastric cancer samples reported that no alteration in expression in the transition from non-neoplastic to gastric primary malignant tissues was identified for MCT1. MCT1 was not associated with TNM stage, and prognostic significance was not assessed [11]. A Korean study (*n* = 45) reported that there was no significant difference in MCT1 expression measured by real-time PCR between gastric cancer tissue and normal gastric tissue samples. However, in 48 patient-derived cells collected from malignant ascites, MCT1 was overexpressed compared with normal tissue or primary gastric cancer tissue. The downregulation of MCTs reduced cancer cell proliferation as well as lactate uptake in a subset of gastric cancer cell lines that overexpressed MCTs. Association with prognosis and clinical pathological variables were not assessed [4]. In our study, high cytoplasmic MCT1 expression was associated statistically significantly with higher T-class (*p* = 0.020), but it was not associated with the other factors. The findings of the present study indicate that acidifying tumor microenvironment tumors become more invasive. Taken together, high MCT1 expression may associate with gastric cancer progression, but not prognosis.

The role of MCT4 in GC has been evaluated in four previous studies with smaller sample sizes. Two Chinese studies (*n* = 113) [3] and (*n* = 143) [10] with immunohistochemically stained gastric cancer samples analyzed both stromal and tumor cell MCT4 expression. Both studies reported that immunohistochemically stained tumor cells expressing MCT4 did not associate with patient clinical parameters and had no prognostic value. However, high MCT4 expression in stromal cells displayed a significantly shorter overall survival, disease-free survival, and was associated with advanced TNM stage. It is good to notice that in the latter study, [10] MCT4 protein staining in tumor cells was reported to be weak or clean, which may affect the interpretation of the results. On the contrary, a Portuguese study (*n* = 190) with immunohistochemically stained gastric reported that a significant decrease in MCT4 expression was observed in the transition and also positive association with lymph node metastasis. They found a significant association with MCT4 expression and Lauren’s intestinal type but not TNM stage. The prognostic significance of MCT4 was not assessed [11]. A Korean study (*n* = 45) reported that MCT4 expression measured by real-time PCR was significantly increased in primary CG tissue compared with normal gastric tissue, and in 48 patient-derived cells collected from malignant ascites, MCT4 was overexpressed compared with normal or primary GC tissue. The downregulation of MCTs reduced cancer cell proliferation as well as lactate uptake in a subset of GC cell lines that overexpressed MCTs. Association with prognosis and clinical pathological variables were not assessed [4]. As Pinheiro et al. [11] highlighted in their study, gastric mucosa is permanently under acidic conditions, and MCT4 is already highly expressed in normal gastric cells. In the present study, MCT4 expression in tumor cells is associated with positive lymph node status, but not prognosis. Taken together, high MCT4 expression might be associated with gastric cancer progression, but not prognosis.

The role of MTCO1 in GC has been evaluated in one previous study with a smaller sample size. A Chinese study (*n* = 42) reported that MTCO1 expression measured by real-time PCR was significantly increased in primary gastric cancerous tissue compared to normal gastric tissue. The expression level was higher in Laurén’s diffuse gastric cancers compared to intestinal type cancers. MTCO1 expression was not associated with stage or lymph-node metastasis [17]. This is inconsistent with our study. The sample size is much smaller compared to our study, and the method used is different. In the present study, low MTCO1 associated with positive distant metastases. Taken together, altered mitochondrial metabolism might be involved in gastric tumorigenesis, but further research on gastric cancer is recommended.

## 5. Conclusions

Previous studies have highlighted lactate transporters as potential therapeutic targets, but the expression of MCTs varies between different cancers. The high expression of MCT1 and MCT4 and low expression of MTCO1 in tumor cells may be associated with poor prognostic variables, but they seem to have no independent prognostic significance in surgically treated gastric adenocarcinoma. Further investigations are required to elucidate the function of lactate metabolism in gastric cancer.

## Figures and Tables

**Figure 1 cancers-13-02142-f001:**
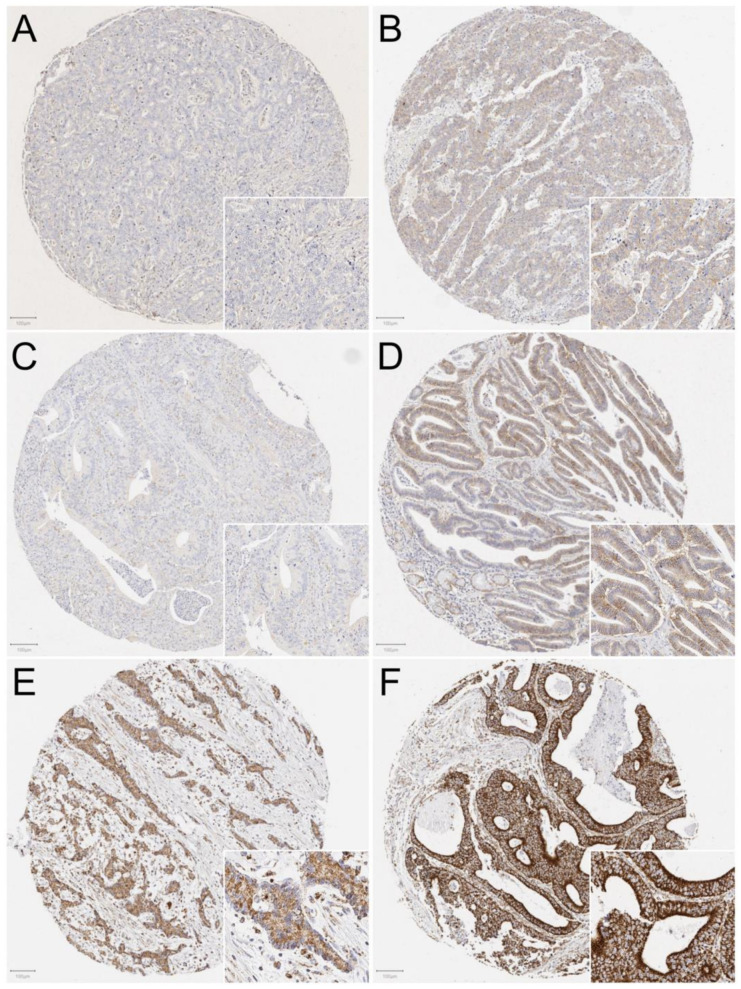
Representative images of immunostainings in the intestinal-type gastric adenocarcinoma. Low MCT1 expression (**A**), high MCT1 expression (**B**), low MCT4 expression (**C**), high MCT4 expression (**D**), low MTCO1 expression (**E**), and high MTCO1 expression (**F**).

**Table 1 cancers-13-02142-t001:** Associations between MCT1 expression and clinicopathological variables in 560 surgically resected patients with gastric adenocarcinoma.

Variable		MCT1		Total	*p*-Value
		Low	High		
Year of surgery					
	≥2000	136 (44.6%)	111 (43.5%)	247/560	0.864
	<2000	169 (55.4%)	144 (56.5%)	313/560	
Age					
	<69	163 (53.4%)	120 (47.1%)	283/560	0.149
	≥69	142 (46.6%)	135 (52.1%)	277/560	
Sex					
	Male	191 (62.6%)	150 (58.8%)	341/560	0.385
	Female	114 (37.4%)	105 (41.2%)	219/560	
T					
	T_1 + 2_	103 (33.8%)	63 (24.7%)	166/560	**0.020**
	T_3 + 4_	202 (66.2%)	192 (75.3%)	394/560	
Lymph nodes					
	negative	163 (53.4%)	132 (51.8%)	295/560	0.734
	positive	142 (46.6%)	123 (48.2%)	265/560	
Organ metastases					
	negative	292 (95.7%)	235 (92.2%)	527/560	0.104
	positive	13 (4.3%)	20 (7.8%)	33/560	
Stage					
	I + II	186 (61.0%)	155 (60.8%)	341/560	0.515
	III + IV	119 (39.0%)	100 (39.2%)	219/560	
Histological grade in intestinal type					
	I	99 (62.3%)	79 (62.7%)	178/285	1.000
	II + III	60 (37.7%)	47 (37.3%)	107/285	
Lauren					
	Intestinal	159 (52.1%)	126 (49.4%)	285/560	0.790
	Diffuse	137 (44.9%)	120 (47.1%)	257/560	
	Other *	9 (3.0%)	9 (3.5%)	18/560	
Perioperative chemotherapy					
	Yes	10 (3.3%)	12 (4.7%)	538/560	0.513
	No	295 (96.7%)	243 (94.3%)	22/560	
Radicality of resection					
	R0	234 (76.7%)	183 (71.8%)	417/560	0.206
	R1 or R2	71 (23.3%)	72 (28.2%)	143/560	

Other *: mixed, no classified; Statistically significant values are bolded.

**Table 2 cancers-13-02142-t002:** Associations between MCT4 expression and clinicopathological variables in 558 surgically resected patients with gastric adenocarcinoma.

Variable		MCT4		Total	*p*-Value
		Low	High		
Year of surgery					
	≥2000	100 (34.5%)	148 (55.2%)	248/558	**0.000**
	<2000	190 (65.5%)	120 (44.8%)	310/558	
Age					
	<69	152 (52.4%)	129 (48.1%)	281/558	0.351
	≥69	138 (47.6%)	139 (51.9%)	277/558	
Sex					
	Male	173 (59.7%)	168 (62.7%)	341/558	0.487
	Female	117 (40.3%)	100 (37.3%)	217/558	
T					
	T_1 + 2_	96 (33.1%)	68 (25.4%)	164/558	0.051
	T_3 + 4_	194 (66.9%)	200 (74.6%)	394/558	
Lymph nodes					
	negative	169 (58.3%)	124 (46.3%)	293/558	**0.005**
	positive	121 (41.7%)	144 (53.7%)	265/558	
Organ metastases					
	negative	273 (94.1%)	252 (94.0%)	525/558	1.000
	positive	17 (5.9%)	16 (6.0%)	33/558	
Stage					
	I + II	180 (62.1%)	159 (59.3%)	339/558	0.282
	III + IV	110 (37.9%)	109 (40.7%)	219/558	
Histological grade in intestinal type					
	I	80 (66.7%)	96 (58.9%)	176/283	0.215
	II + III	40 (33.3%)	67 (41.1%)	107/283	
Lauren					
	Intestinal	120 (41.4%)	163 (60.8%)	283/558	**0.000**
	Diffuse	159 (54.8%)	97 (36.2%)	256/558	
	Other *	11 (3.8%)	8 (3.0%)	19/558	
Perioperative chemotherapy					
	Yes	5 (1.7%)	17 (6.3%)	22/558	**0.008**
	No	285 (98.3%)	251 (93.7%)	536/558	
Radicality of resection					
	R0	212 (73.1%)	202 (75.4%)		0.562
	R1 or R2	78 (26.9%)	66 (24.6%)		

Other *: mixed, no classified; Statistically significant values are bolded.

**Table 3 cancers-13-02142-t003:** Associations between MTCO1 expression and clinicopathological variables in 562 surgically resected patients with gastric adenocarcinoma.

Variable		MTCO1		Total	*p*-Value
		Low	High		
Year of surgery					
	≥2000	149 (49.8%)	99 (37.6%)	248/562	**0.004**
	<2000	150 (50.2%)	164 (62.4%)	314/562	
Age					
	<69	153 (51.2%)	131 (49.8%)	284/562	0.800
	≥69	146 (48.8%)	132 (50.2%)	278/562	
Sex					
	Male	173 (57.9%)	169 (64.3%)	342/562	0.141
	Female	126 (42.1%)	94 (35.7%)	220/562	
T					
	T_1 + 2_	94 (31.4%)	74 (28.1%)	168/562	0.407
	T_3 + 4_	205 (68.6%)	189 (71.9%)	394/562	
Lymph nodes					
	negative	160 (53.5%)	136 (51.7%)	296/562	0.673
	positive	139 (46.5%)	127 (48.3%)	266/562	
Organ metastases					
	negative	275 (92.0%)	254 (96.6%)	529/562	**0.030**
	positive	24 (8.0%)	9 (3.4%)	33/562	
Stage					
	I + II	179 (59.9%)	164 (62.4%)	343/562	0.603
	III + IV	120 (40.1%)	99 (37.6%)	219/562	
Histological grade in intestinal type					
	I	83 (60.1%)	95 (64.6%)	178/285	0.464
	II + III	55 (39.9%)	52 (35.4%)	107/285	
Lauren					
	Intestinal	138 (46.2%)	147 (55.9%)	285/562	**0.044**
	Diffuse	148 (49.5%)	110 (41.8%)	258/562	
	Other *	13 (4.3%)	6 (2.3%)	19/562	
Perioperative chemotherapy					
	Yes	15 (5.0%)	6 (2.3%)	541/562	0.118
	No	284 (95.0%)	257 (97.7%)	21/562	
Radicality of resection					
	R0	219 (73.2%)	199 (75.7%)	418/562	0.288
	R1 or R2	80 (26.8%)	64 (24.3%)	144/562	

Other *: mixed, no classified; Statistically significant values are bolded.

**Table 4 cancers-13-02142-t004:** Univariable and multivariable analysis of MCT1, MCT4, and MTCO1 expression and prognosis in 568 patients with gastric adenocarcinoma.

	MCT1		MCT4		MTCO1	
	Number of Patients	High MCT1 HR (95% CI)	Number of Patients	High MCT4 HR (95% CI)	Number of Patients	High MTCO1 HR (95% CI)
5-year survival						
All patients (Crude)	560	1.06 (0.86–1.30)	558	1.03 (0.84–1.26)	562	1.08 (0.88–1.32)
All patients (Adjusted) ^a^	560	1.03 (0.84–1.26)	558	1.07 (0.86–1.33)	562	1.10 (0.90–1.35)
*Subgroup analysis*						
Intestinal type (Crude)	285	1.13 (0.85–1.50)	283	1.14 (0.86–1.52)	285	0.91 (0.69–1.21)
Intestinal type (Adjusted) ^b^	285	0.97 (0.72–1.30)	283	1.17 (0.87–1.58)	285	0.94 (0.71–1.25)
Diffuse type (Crude)	257	1.04 (0.77–1.41)	256	0.92 (0.67–1.26)	258	1.28 (0.95–1.74)
Diffuse type (Adjusted) ^c^	257	1.09 (0.80–1.47)	256	0.93 (0.67–1.30)	258	1.30 (0.96–1.76)
Overall survival						
All patients (Crude)	560	1.11 (0.92–1.33)	558	1.09 (0.91–1.31)	562	1.08 (0.90–1.29)
All patients (Adjusted) ^a^	560	1.07 (0.89–1.28)	558	1.08 (0.89–1.31)	562	1.12 (0.93–1.35)
*Subgroup analysis*						
Intestinal type (Crude)	285	1.24 (0.96–1.59)	283	1.12 (0.87–1.44)	285	0.89 (0.69–1.15)
Intestinal type (Adjusted) ^b^	285	1.07 (0.82–1.39)	283	1.14 (0.87–1.48)	285	0.96 (0.75–1.24)
Diffuse type (Crude)	257	1.04 (0.79–1.36)	256	1.00 (0.75–1.34)	258	1.26 (0.96–1.66)
Diffuse type (Adjusted) ^c^	257	1.10 (0.83–1.46)	256	0.97 (0.72–1.31)	258	1.27 (0.96–1.67)

MCT1, MCT4 and MTCO1: Low expression HR (95% CI) 1.00 (Reference); a Adjusted for year of diagnosis, age, sex, tumor stage, Lauren classification, perioperative chemotherapy, and radical resection; b Adjusted for year of diagnosis, age, sex, tumor stage, tumor grade, perioperative chemotherapy, and radical resection; c Adjusted for year of diagnosis, age, sex, tumor stage, perioperative chemotherapy, and radical resection.

## Data Availability

The data that support the findings of this study are available from the corresponding author upon reasonable request. Case-by-case permissions from the data owners (Statistics Finland, Northern Finland Biobank Borealis) are required for sharing the data.

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
