# Peer review of "Monocarboxylate Transporters 1 and 4 and MTCO1 in Gastric Cancer"

_cancers, 2021, doi:10.3390/cancers13092142_

Round 1
Reviewer 1 Report
The paper studies the expression of MCT1, MCT4 and MTCO1 59 and their association with clinical-pathological features and prognostic significance in a wide cohort of gastric cancer patients. The topic is interesting and important. The series analyzed is the most extensive described in the literature and offers significant value to the retrospective study. It is correctly conducted, well written and the conclusions are very interesting and open to further and new researches, well contextualized in the Special Issue "The Biomarkers for the Diagnosis and Prognosis in Cancer". Although not my elective area of expertise, I have appreciated the research, as the topic is elaborated with precision and clarity, with appropriate bibliographic references. The figures are also very pleasant and didactic.
In my opinion, the paper appears suitable for publication on Cancers.
Author Response
We kindly thank for the favorable overall evaluation and the positive comments.
Reviewer 2 Report
Dear Authors,
these are several suggestions that have raised while reviewing your manuscript:
- I would recommend checking this manuscript once again in terms of English since I have detected several either minor or major mistakes that should be corrected by the Authors
- In the abstract - once you have used MCT while in other lines it is 'MTC'. Please unify it here as well as in further lines in the text.
- I would recommend to add a strong argument WHY the study is important. You have mentioned that MCTs could not be used as a prognostic factor thus why its over expression is clinically important? Or important in terms of other studies? What about the potential utility of MCTs as diagnosis factor of gastric cancer? It would be beneficial to think also in terms of these aspects and add it in the text.
- Line 39 - regarding reference 1. It is quite old reference in terms of the dynamically changing statistics with respect to gastric cancer epidemiology. Please provide most recent references - e.g. Machlowska et al. 2020 (10.3390/ijms21114012).
- Please correct the references according to the restrictions provided in the 'Author guidelines'.
- Line 49 - could you mention in what MCTs play a role in the development of drug resistance?
- Line 50 - the Authors have mentioned about 'a variety of cancers' while the main topic of this work is related to gastric cancer. I do not claim that this sentence is redundant - however, you should add more information about the gastric cancer specifically.
- Line 61 - I would recommend omitting the word 'large' as it is quite subjective. In fact your study group is representative however, larger are also seen in numerous studies.
- Line 92 - a space is missing after '20' and before 'min'
- Line 137 - I would rather put this information first in the text. The total number of patients first, then the age and gender etc..
- Line 140 - could you indicate how many patients undergo chemotherapy and how many - chemoradiotherapy? Just separate these two
- Table 2 - it is not very clear for the readers. I would recommend narrowing down the very first column and it will be easier for readers to read
- Line 202 - could you provide proper references for this sentence?
Good luck with you further research
A Reviewer
Author Response
Response: We thank you for the favorable overall evaluation and the constructive comments, which have led to improvements in the manuscript. The pages and lines changed slightly as revisions were added into the text. Below we provide point-to-point answers to the questions provided by the referee. The changes in the revised manuscript are highlighted in yellow.
Reviewer #2 comment No. 1 I would recommend checking this manuscript once again in terms of English since I have detected several either minor or major mistakes that should be corrected by the Authors
- Revisions: We corrected the typographical error of word “parametres” to “parameters” on page 2, line 62 and on page 1, lines 15-16; word “perticipants” to “participants” on page 2, line 83; word “preoperative” to “perioperative” on page 4, line 144; word “MCT” to “MCT1” on page 4, line 145; typographic errors on page 7-8, table 4. We also made some stylistic and linguistic edits.
Reviewer #2 comment No. 2 In the abstract - once you have used MCT while in other lines it is 'MTC'. Please unify it here as well as in further lines in the text.
- Response: We thank for pointing typographic errors.
- Revisions: We corrected the word “MTC” to “MCT” on page 5, line 159 and on page 6, table 2.
Reviewer #2 comment No. 3 I would recommend to add a strong argument WHY the study is important. You have mentioned that MCTs could not be used as a prognostic factor thus why its over expression is clinically important? Or important in terms of other studies? What about the potential utility of MCTs as diagnosis factor of gastric cancer? It would be beneficial to think also in terms of these aspects and add it in the text.
- Response: Thank you for these good observations. The exact role of MCTs in gastric cancer is not yet fully known, as previous studies are contradictory. If MCTs are considered potential drug targets, their expression in gastric cancer is required. However, any prognostic significance for MCT might not be necessary, if a clear mechanism for a drug exists, as is the case for MCTs. However, higher expression of MCTs was associated with poor prognostic variables, supporting the potential of MCT inhibitors in the future. For diagnostic significance, no comparison to normal mucosa, or any diagnostic series was or could be done. Therefore, no conclusions on diagnosis of gastric cancer and MCTs could be drawn in this study.
- Revisions: We changed the order of the sentences and emphasized the importance of this study on page 2, lines 50-57.
Reviewer #2 comment No. 4 Line 39 - regarding reference 1. It is quite old reference in terms of the dynamically changing statistics with respect to gastric cancer epidemiology. Please provide most recent references - e.g. Machlowska et al. 2020 (10.3390/ijms21114012).
- Response: We thank for this important suggestion. We updated the incidence and mortality statistics to the most comprehensive and up-to-date original publications of GLOBOCAN 2020 and GBD 2017 estimates, as the cancer data for GBD 2019 is yet to be published. The suggested reference in IJMS is a review article, and refers to GLOBOCAN data published in 2008 for incidence and mortality. Therefore, we feel that the references in the current version are the most up-to-date.
- Revisions: We exchanged statistical data to match the data reported in the following articles: Sung H, Ferlay J, Siegel RL, Laversanne M, Soerjomataram I, Jemal A, et al. Global cancer statistics 2020: GLOBOCAN estimates of incidence and mortality worldwide for 36 cancers in 185 countries. CA. Cancer J. Clin. 2021, 0, 1-41, doi:10.3322/caac.21660. and Global, regional, and national cancer incidence, mortality, years of life lost, years lived with disability, and disability-adjusted life-years for 29 cancer groups, 1990 to 2017. JAMA Oncol, 2019, 5(12):1749-1768, doi:10.1001/jamaoncol.2019.2996. on page 1, lines 39-40.
Reviewer #2 comment No. 5 Please correct the references according to the restrictions provided in the 'Author guidelines'.
- Revisions: We corrected the references according to the journal guidelines.
Reviewer #2 comment No. 6 Line 49 - could you mention in what MCTs play a role in the development of drug resistance?
- Response: Both MCT1 and MCT4 are reported to be associated with the development of drug resistance, mainly by acidifying tumor microenvironment.
- Revisions: Since there was only one reference after this sentence, and only MCT1 involvement was discussed in that study, we added another reference on page 2, line 50.
Reviewer #2 comment No. 7 Line 50 - the Authors have mentioned about 'a variety of cancers' while the main topic of this work is related to gastric cancer. I do not claim that this sentence is redundant - however, you should add more information about the gastric cancer specifically.
- Revisions: We changed the sentence structure on page 2, lines 50-57, so that MCTs in gastric cancer is the main point.
Reviewer #2 comment No. 8 Line 61 - I would recommend omitting the word 'large' as it is quite subjective. In fact your study group is representative however, larger are also seen in numerous studies.
- Revisions: We omitted the word “large” from the text on page 2, line 63 and from the simple summary on page 1, line 16.
Reviewer #2 comment No. 9 Line 92 - a space is missing after '20' and before 'min'
- Revisions: We corrected this typographic error, and also other similar space-errors on page 2 and 3.
Reviewer #2 comment No. 10 Line 137 - I would rather put this information first in the text. The total number of patients first, then the age and gender etc..
- Revisions: We added the total number of patients first on page 3, line 138.
Reviewer #2 comment No. 11 Line 140 - could you indicate how many patients undergo chemotherapy and how many - chemoradiotherapy? Just separate these two
- Response: We are sorry for this error. All patients received chemotherapy, and none received radiotherapy.
- Revisions: Word “chemo(radio)therapy” was changed to “chemotherapy” on page 4, line 144.
Reviewer #2 comment No. 12 Table 2 - it is not very clear for the readers. I would recommend narrowing down the very first column and it will be easier for readers to read
- Response: Thank you for this note. We are afraid this issue is related to the typesetting of the manuscript by the editorial office. We however agree that narrow tables would be better.
Reviewer #2 comment No. 13 Line 202 - could you provide proper references for this sentence?
- Response: The sample size of previous studies is varying from 45 to 190 patients, which is why we have described this study “much larger”.
- Revisions: We added references to support this sentence on page 8, line 207. We omitted the word “much” from the text (“much larger” to “larger”) on line 206.
Reviewer 3 Report
In this study Eskuri et al evaluated the expression of lactate transporters MCT1 and MCT4 as well as the mitochondria cytochrome c oxidase (MTCO1) in gastric tumors, thought a retrospective cohort study. Statistical analysis demonstrated that MCT4, but not MCT1, increased expression is correlated with tumor dissemination. On the opposite, low levels of MTCO1 were observed in tumors from patients who presented distant metastases. However, none of the markers demonstrated to be associated with prognosis. The relevance of this study stands in the big size of the cohort and the absence of a selection bias, compared to previous studies involving MCT’s transporter family and MTCO1. The article is clearly written and well structured.
Page 2, line 48: Please confirm the use of “the glycolysis” in the sentence.
Page 3, line 137: Please add definition of R0 and R1/2 resection.
Page 4, Figure 1. Do the authors have images of all the tumor cut area? Would add more value to see the homogeneity or heterogenicity of the different markers depending on the zone of the tumor (margins versus core). The different location of the staining would give some indications of why variations in these markers are more correlated with dissemination and progression.
Author Response
Response: We kindly thank you for the favorable overall evaluation and the constructive comments, which have led to improvements in the manuscript. The pages and lines changed slightly as revisions were added into the text. Below we provide point-to-point answers to the questions provided by the referee. The changes in the revised manuscript are highlighted in yellow.
Response to Reviewer #3 comment No. 1 Page 2, line 48: Please confirm the use of “the glycolysis” in the sentence.
- Revision: We omitted article “the” from the text on page 2, line 48.
Response to Reviewer #3 comment No. 2 Page 3, line 137: Please add definition of R0 and R1/2 resection.
- Response: We thank you for this important note.
- Revisions: We added “R0 resection was defined as no cancer cells seen microscopically at the tumor border. R1/2 resection was defined as tumor growth on the border of the resected specimen, or macroscopic residual disease.” on page 3, lines 129-131.
Response to Reviewer #3 comment No. 3 Page 4, Figure 1. Do the authors have images of all the tumor cut area? Would add more value to see the homogeneity or heterogenicity of the different markers depending on the zone of the tumor (margins versuscore). The different location of the staining would give some indications of why variations in these markers are more correlated with dissemination and progression.
- Response: Thank you for the good comment. Unfortunately, we do not have stained full sections of the tumors, but instead used tissue microarrays with 1mm cores. A larger area can be visualized with the complete cores, compared to the small photomicrographs that were used in the previous version of the manuscript.
- Revisions: We have changed the images to photomicrographs of full cores with higher-magnification inserts, so that the heterogeneity of the markers can be seen better.
Round 2
Reviewer 2 Report
Dear Authors,
thank you for correcting your manuscript according to my suggestions and comments. I suppose that the manuscript has been significantly improved and at this point I have no further comments.
Good luck with your further scientific research and work!